# R4D-planes: Remapping Planes For Novel View Synthesis and Self-Supervised Decoupling of Monocular Videos

## ABSTRACT

The tasks of view synthesis and decoupling dynamic objects from the static environment for monocular scenes are both long-standing challenges in CV and CG. Most of the previous NeRF-based methods rely on implicit representation, which require additional supervision and training time. Later, various explicit representations like multi-planes or 3D gaussian splatting have been extended and applied to the task of novel view synthesis for dynamic scenes. They introduce an additional time dimension or a deformation field into the original representation to encode dynamics. Due to the effective explicit representations, these methods greatly reduce the time consumption, but still fail to achieve high rendering quality in some scenes, especially for some real scenes. For the latter decoupling problem, previous neural radiation field methods require frequent tuning of the relevant parameters for different scenes, which is very inconvenient for practical use. We consider above problems and propose a new representation of dynamic scenes based on tensor decomposition, which we call R4D-planes. The key to our method is remapping, which compensates for the shortcomings of the plane structure by fusing space-time information and remapping to new indexes. Furthermore, we implement a new decoupling structure, which can efficiently decouple dynamic and static scenes in a self-supervised manner. Experimental results show our method achieves better rendering quality and training efficiency in both view synthesis and decoupling tasks for monocular scenes.

## CCS CONCEPTS

• **Computing methodologies** → **Computer graphics**; **Computer vision**.

## KEYWORDS

Neural Radiance Field, View Synthesis, Self-supervised Decoupling, Monocular Video

## 1 INTRODUCTION

How to accurately reconstruct and render a 3D scene is a long-standing and important issue in computer vision and computer graphics, related technologies can be widely applied in fields such as augmented reality/virtual reality (AR/VR), movies, games, and others. Recently, with the development of neural rendering, many methods based on Neural Radiance Field(NeRF)[16] have emerged,

demonstrating impressive performance in both reconstruction accuracy and rendering quality with only multi-view images and corresponding camera position information. However, it cannot be ignored that data in the form of dynamic monocular video is the most common and easily accessible form for a variety of real-life situations with practical capture conditions.

Many current approaches for dynamic scenarios extend on the basis of static implicit representations. However, implicit representations rely on large MLPs[13, 14, 18, 19, 21, 29, 33], which typically incur significant training and inference time. At the same time, the nature of MLP causes the radiance field to be unable to effectively capture the high frequency signals in the scene.

In order to improve the rendering efficiency of NeRF, researchers have proposed various improvement strategies, such as improved sampling strategies or pre-computation . Later, a number of different explicit representations have been explored, including voxels[2, 17, 25], low-rank tensor decomposition[4, 10], 3D Gaussians[12], etc. These representations lead to breakthroughs in rendering efficiency, some even enabling real-time rendering. In terms of quality improvement, Mip-NeRF effectively reduces aliasing artifacts and dramatically improves the ability to express fine details, its core idea has subsequently been applied in related explicit representations, including voxels[2], tri-planes[10], and 3D Gaussians.

These representations are quickly extended to dynamic scenes, unlike implicit representations, explicit structures usually greatly reduce time consumption, but increase memory overhead. Most of them use hybrid methods[3, 5, 6, 9], which typically extract features using explicit structures and decode them using a small MLP. Representations in these methods are usually in the form of voxel mesh or its decomposition, which still presents some inconveniences when dealing with dynamic scenes. For example, simply extending the voxel grid to four dimensions will lead unaffordable memory consumption, while this problem can be alleviated by employing decomposed representations, tensor decomposition structures need to be low-rank, which are often difficult to optimize due to lack of constraints in dynamic monocular scenes. Another methods follow the pipeline of 3D Gaussian Splatting(3DGS), using 4D gaussians[31] or deformable 3D gaussians[27, 30] to represent the dynamics, but cases show they fail to modeling large motions and rapid scene changes. Therefore, there are still some problems in monocular scenes. Using additional supervision, such as optical flow[14, 15], depth[1, 14, 15], etc. would be of great help, but in practical situations those are often not easily obtained.

We consider the above problems and make further improvements to the multi-plane representation based on tensor decomposition. In general, the tensor decomposition structure is sparse and low-rank. However, optimizing such a low-rank grid is not straightforward, studies[8, 32] have shown that optimization can be stable only when the frame of representation aligns with the scene or signal structure. On the other hand, for data in the form of monocular

videos, there must be an information interaction mechanism between the video frames to maintain the consistency of geometry. Previous multi-plane methods[3, 6] ignored above factors, resulting in poor rendering quality and geometry in monocular scenes.

In this paper, we propose a new representation of dynamic scenes based on tensor decomposition. We focus on exploiting spatial location and temporal information by remapping the 4D coordinates through the remapping net, which allows for a better fusion of the spatio-temporal information with the explicit planes. By remapping, we achieve a separation of the scene from the explicit representation (position of sampling point in the scene is not same as explicit representation), which makes the alignment condition non-essential and reduces the difficulty of optimization. Except for the task of view synthesis, we also designed a new structure for the self-supervised decoupling task, greatly reducing the time cost, and achieving promising performance in decoupling and synthesis results.

In summary, the main contributions of this work are as follows:

• A new representation of dynamic scenes, which extracts the feature by the new 4D coordinates after remapping, can overcome the shortcomings of previous methods and achieve realistic rendering results.

• A new explicit structure for dynamic and static self-supervised decoupling and its optimization strategies, showing better efficiency and decoupling results.

• Experiments show that our method is simple and effective, bridges the gap of previous multi-plane methods on monocular data, and achieves high-quality results on multiple datasets without additional supervision.

## 2 RELATED WORK

### 2.1 Explicit Representations for Dynamic Scenes

Explicit representations have been widely applied to dynamic scenes, voxel is an effective way to do this. Tineuvox[5] uses a deformation net and replaces the MLP for static canonical space with a voxel grid. NeRFPlayer[24] shows that the time dimension cannot simply be introduced, which would make the memory consumption of long videos unaffordable. Temporal interpolation NeRF[20] suggests modeling different time periods separately and then performing temporal interpolation to get the results, but this does not seem to work for all scenarios. The VM decomposition in TensoRF[4] makes voxel representation much less consuming, and K-planes, Hexplane, and Tensor4D[3, 6, 23] decompose dynamic scenes based on this low-rank decomposition with 2D tensors. HumanRF[11] defines 4D feature grids as a decomposition of four 3D and four 1D feature grids and uses adaptive temporal partitioning to model human motion. However, without additional supervision, for monocular video, especially in real-world scenes, although these decomposition methods can cope with some topology changes and reduce the time overhead, the results are not comparable to previous methods, thus they are more likely to be used in multi-view settings. Recently, due to the excellent performance of 3DGS[12], some recent work has extended it to dynamic scenes as well. Similar to NeRFs, some replace the 3D gaussians with 4D gaussians and others introduce a

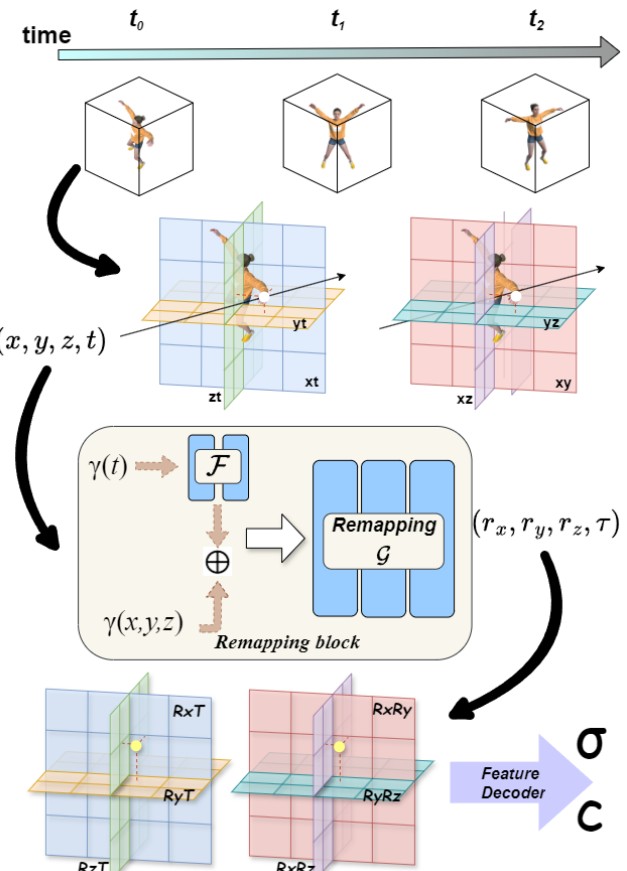

**Figure 1: Overview of the remapping approach for view synthesis task, instead of using $(x, y, z, t)$ for feature extraction in the structure, we remapped the sampled $xyzt$ and use the new coordinate $(r_x, r_y, r_z, \tau)$ for feature extraction. The features are then decoded into color $c$ and density $\sigma$. See supplementary material for detailed net structure.**

deformation field. However, current works seem to have difficulty in modeling large motions and rapid scene changes.

### 2.2 Decoupled 3D Representation

To reduce the complexity of dynamic scenes, there are already some approaches that tend to model dynamic objects and static backgrounds separately, but they aim to improve the quality of the rendering rather than cleanly decoupling the static and dynamic parts at the same time[14]. STaR[33] proposes a framework for self-supervised decoupling and reconstruction of moving objects and scenes, but it is only suitable for a single rigidly moving object and requires multi-view videos. Using HyperNeRF[19] and NeRF[16] to model the dynamic foreground and static background respectively, $D^2$NeRF[28] achieves self-supervised decoupling of dynamic and static objects and clean rendering results of both through a series of loss functions and hyperparameters. However, apart from the time it takes, the noise in static rendering caused by the occlusion

of dynamic objects cannot be ignored. In addition, frequent tuning of the hyper-parameters for different scenes definitely makes the application more difficult. SUDS[26] advantages the method in $D^2$NeRF and applies it to large-scale scenarios, building a large-scale dynamic NeRF supervised by signals such as radar, optical flow, etc., and enabling downstream tasks such as viewpoint synthesis, 3D scene flow estimation, etc. RobustNeRF[22] models distractors in training data as outliers of an optimization problem, and removes outliers from the scene to get a clean static rendering result, but the distractor is the same or different target in frames, rather than an object moving continuously over time.

## 3  METHOD

In this section, we first review the general architecture of multi-plane representations(Sec.3.1). We then describe the implementation of the remapping method and illustrate the representation for view synthesis task(Sec.3.2). We show the newly proposed structure for the dynamic and static decoupling task, which is based on tensor decomposition(Sec.3.3). Finally, we show the optimization details in experiments(Sec.3.4).

### 3.1  Review of Multi-plane Representation

Most of the neural radiation fields based on explicit representations employ the form of hybrid neural fields i.e., for sampled points in space, interpolation is performed at the corresponding positions in the explicit representation to obtain the latent features, then they will be decoded by a small MLP to obtain the attributes of the radiance field, color or density. This hybrid structure gives a unique advantage to the neural radiance field, achieving good performance in terms of quality and efficiency of rendering.

The multi-plane representation also follows the hybrid structure described above, with a decomposition of the tensor representing the latent feature volume in it. The original dense grid is decomposed into a series of low-dimensional factors, which greatly reduces memory consumption and makes the structure more compact and efficient. The sampled points in the space are projected into the corresponding low-dimensional coordinate space, and the latent features are obtained by interpolating over the projected point positions,then feature aggregation such as concatenation, multiplication, or addition will be performed to obtain the actual latent features $D$ . The process can be expressed as:

$$D = \text{Aggr}\big(\big[\text{Interp}_{F_1}(\text{Proj}_1(\boldsymbol{p}))\big], \dots, \\ \big[\text{Interp}_{F_F}(\text{Proj}_F(\boldsymbol{p}))\big]\big). \tag{1}$$

where Aggr denotes the aggregation process, Proj denotes projection, Interp denotes interpolation, $p$ is the position of the sampled points, $F_1, \dots F_F$ denotes the corresponding low-dimensional factors.

For the 4D representation of a dynamic scene, it is usually decomposed into the form of Hexplane[3]. It uses six learnable parameter planes to encode temporal and spatial information. For a sampled point $(x, y, z, t)$ in space at a given moment, it will be projected onto the six planes including XY-ZT, YZ-XT, XZ-YT and the feature

vectors of the point will be computed by the following form:

$$D = \sum_{r=1}^{R_1} M_r^{XY} \odot M_r^{ZT} \odot v_r^1 + \sum_{r=1}^{R_2} M_r^{XZ} \odot M_r^{TY} \odot v_r^2 \\ + \sum_{r=1}^{R_3} M_r^{YZ} \odot M_r^{XT} \odot v_r^3, \tag{2}$$

where $M_r^{AB} \in \mathbb{R}^{AB}$ is the feature in the corresponding plane after interpolation, $D$ is the feature vector that will be decoded.

### 3.2  R4D-plane Representation for Monocular data

In this section, we present our framework for dynamic monocular reconstruction. Previous representation based on tensor decomposition, such as Hexplane[3], have shown limitations in monocular settings. The structure lacks an information sharing mechanism between frames, cannot exploit the information in different frames to constrain the object reconstruction. On the other hand, due to lack of constraints in monocular dynamic scenes, optimizing the tensor to be low-rank is hard. In fact, in dynamic scenes, the color and density can change abruptly in space and time, Total Variational (TV) loss cannot effectively impose constraints in this case, instead it makes neighboring feature in the structure similar. Furthermore, representations like voxel or tri-plane using $xyzt$ coordinates are difficult to capture high-frequency signals due to interpolation at a limited resolution. Previous work use multi-resolution in space to alleviate this problem, but our work takes a different approach.

As shown in Fig.1, when a point $p$ in a ray with direction $(\theta, \phi)$ is sampled at $(x, y, z, t)$. The time t will be first encoded by a small MLP $\mathcal{F}$ and then concatenated with the position encoding of $(x, y, z)$ into the remapping MLP $\mathcal{G}$ to obtain the new coordinate $(r_x, r_y, r_z, \tau)$. Among them,$\tau$ is activated by $tanh$, compressing its range to [-1,1]:

$$\mathcal{G}(\gamma(x, y, z), \mathcal{F}(\gamma(t))) = (r_x, r_y, r_z, \tau) \tag{3}$$

where $\gamma$ is the position encoding:

$$\gamma(x) = (sin(2^0 x), cos(2^0 x), \dots, sin(2^{L-1} x), cos(2^{L-1} x)) \tag{4}$$

where $L$ is a hyperparameter which controls frequency of the encoding.

Remapping has two main functions: firstly, it fuses information from time and space, introducing a mechanism for sharing information across frames. Secondly, the interpolation position is also optimized compared to direct indexing using fixed $xyzt$ coordinate, which means that the original samples are reconfigured and have different scales in the new dimension.

Then we project it onto these six planes and use linear interpolation to get the density feature:

$$\mathcal{D} = \mathbf{P}_{\sigma, r_x r_y}^{R_x R_y F} \circ \mathbf{P}_{\sigma, r_z \tau}^{R_z TF} + \mathbf{P}_{\sigma, r_x r_z}^{R_x R_z F} \circ \mathbf{P}_{\sigma, r_y \tau}^{R_y TF} \\ + \mathbf{P}_{\sigma, r_y r_z}^{R_y R_z F} \circ \mathbf{P}_{\sigma, r_x \tau}^{R_x TF} \tag{5}$$

and appearance feature $\mathcal{A}$ :

$$(\mathbf{P}_{c, r_x r_y}^{R_x R_y F} \circ \mathbf{P}_{c, r_z \tau}^{R_z TF}, \mathbf{P}_{c, r_x r_z}^{R_x R_z F} \circ \mathbf{P}_{c, r_y \tau}^{R_y TF}, \mathbf{P}_{c, r_y r_z}^{R_y R_z F} \circ \mathbf{P}_{c, r_x \tau}^{R_x TF}) \tag{6}$$

where $\circ$ is outer product, $\mathbf{P}_{\sigma, r_x r_y}^{R_x R_y F} \in \mathbf{R}^{R_x R_y F}$ is the corresponding feature in each plane. Feature $\mathcal{A}$ and $\mathcal{D}$ are eventually decoded

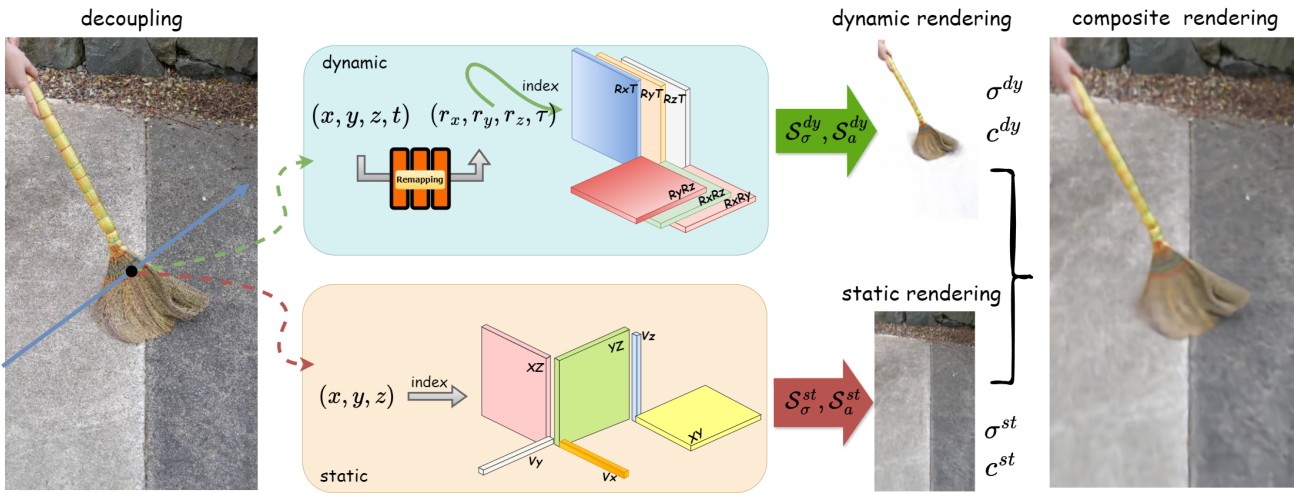

**Figure 2: The structure for decoupling of static and dynamic task. We are the first to decouple the static and dynamic parts for monocular video in a self-supervised manner using explicit representation. We use the remapping representation as the dynamic component, and for the static component, we follow the vm decomposition in TensoRF[4] to represent the static space, since it does not include the time dimension.**

into color and density by a direct sum or a tiny MLP.

**Is remapping block a deformation field?** Although we use an MLP to output a set of values about coordinates, the idea of remapping has no relation to deformation fields. For the method of reconstructing dynamic scenes using a deformation field, the key is to map the sampled point $p$ given a moment($p \in \mathbb{R}^4$) into a canonical space $H$ ($H \in \mathbb{R}^3$), compressing the originally four-dimensional space into a three-dimensional canonical space. This assumes that any point in the originally four-dimensional space must have a counterpart in the canonical space. But for some cases this assumption is not correct.

In contrast, our method is a remapping of the position, a transformation of the entire domain, a mapping from four dimensions to four dimensions, with no loss of information in scenes, and experiments have demonstrated that our method is effective in capturing topological changes, which is difficult for deformation methods.

### 3.3 Decoupling of Static and Dynamic Objects

In the case of dynamic-static decomposition, we follow the above structure as our dynamic representation, the structure is shown in Fig.2. For the static background we use the VM decomposition in TensoRF[4] and set the dimensions of vector and matrix to the same as the dynamic one:

$$\mathcal{D}^{st} = \mathbf{v}_{\sigma,x}^{XF} \circ \mathbf{M}_{\sigma,yz}^{YZF} + \mathbf{v}_{\sigma,y}^{YF} \circ \mathbf{M}_{\sigma,xz}^{XZF} + \mathbf{v}_{\sigma,z}^{ZF} \circ \mathbf{M}_{\sigma,xy}^{XYF} \quad (7)$$

and

$$\mathcal{A}^{st} = (\mathbf{v}_{c,x}^{XF} \circ \mathbf{M}_{c,yz}^{YZF}, \mathbf{v}_{c,y}^{YF} \circ \mathbf{M}_{c,xz}^{XZF}, \mathbf{v}_{c,z}^{ZF} \circ \mathbf{M}_{c,xy}^{XYF}) \quad (8)$$

Static and dynamic components have their own decoders $\mathcal{S}^{st}$ and $\mathcal{S}^{dy}$:

$$\mathcal{S}_{density}^{st}(\mathcal{D}^{st}) = \sigma^{st}, \mathcal{S}_{appearance}^{st}(\mathcal{A}^{st}) = c^{st} \quad (9)$$

For dynamic component:

$$\mathcal{S}_{density}^{dy}(\mathcal{D}) = \sigma^{dy}, \mathcal{S}_{appearance}^{dy}(\mathcal{A}) = c^{dy} \quad (10)$$

then the pixel colour $\hat{C}$ at a given frame is calculated by volume rendering:

$$\hat{C}(\mathbf{r}) = \sum_{i=1}^{N} \mathcal{T}_i \left( \alpha_i^{st} c_i^{st} + \alpha_i^{dy} c_i^{dy} \right) \quad (11)$$

where $\mathcal{T}_i = \exp\left( -\sum_{j=1}^{i-1} \left( \sigma_j^{st} + \sigma_j^{dy} \right) \delta_j \right) \quad (12)$

$$\alpha_i^{st} = 1 - \exp\left( -\sigma_i^{st} \delta_i \right) \quad (13)$$

and $\alpha_i^{dy} = 1 - \exp\left( -\sigma_i^{dy} \delta_i \right) \quad (14)$

In this setup, dynamic and static scenes can naturally be rendered individually, following the body rendering equations:

$$\hat{C}_{dy}(\mathbf{r}) = \sum_{i=1}^{N} \mathcal{T}_i^{dy} \left( \alpha_i^{dy} c_i^{dy} \right) \quad (15)$$

where $\mathcal{T}_i^{dy} = \exp\left( -\sum_{j=1}^{i-1} \left( \sigma_j^{dy} \right) \delta_j \right) \quad (16)$

and

$$\hat{C}_{st}(\mathbf{r}) = \sum_{i=1}^{N} \mathcal{T}_i^{st} \left( \alpha_i^{st} c_i^{st} \right) \quad (17)$$

where $\mathcal{T}_i^{st} = \exp\left( -\sum_{j=1}^{i-1} \left( \sigma_j^{st} \right) \delta_j \right) \quad (18)$

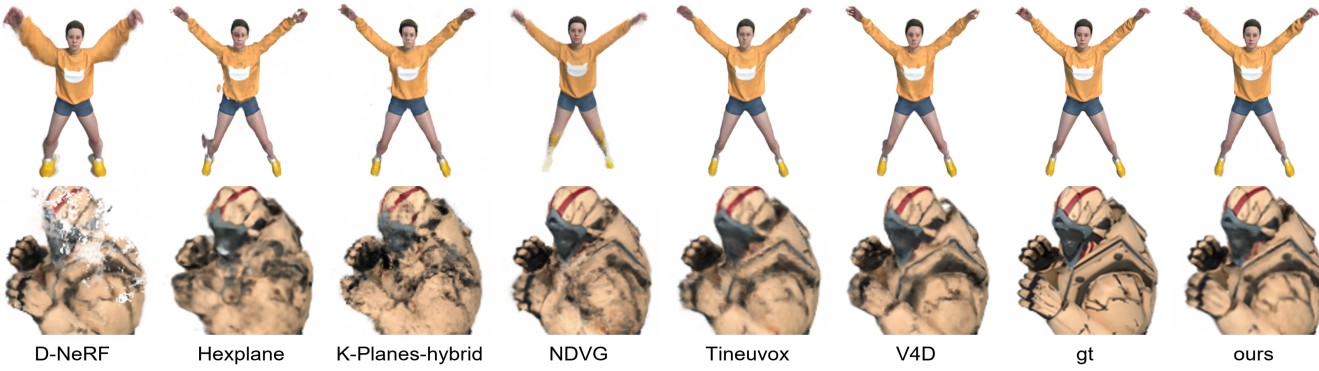

**Figure 3: Qualitative comparison in synthetic scenes**

## 3.4 Optimization Details

**Loss.** For monocular view synthesis, the optimization objective is:

$$\mathcal{L} = \mathcal{L}_c + \lambda_{tv}\mathcal{L}_{tv} \tag{19}$$

$$\mathcal{L}_c(\mathbf{r}) = \|C(\mathbf{r}) - \hat{C}(\mathbf{r})\|_2^2 \tag{20}$$

$$\mathcal{L}_{tv}(\mathbf{P}) = \frac{1}{|C|n^2} \sum_{c,i,j} \left( \left\| \mathbf{P}_c^{i,j} - \mathbf{P}_c^{i-1,j} \right\|_2^2 + \left\| \mathbf{P}_c^{i,j} - \mathbf{P}_c^{i,j-1} \right\|_2^2 \right) \tag{21}$$

where $\mathbf{r}$ is the sampled ray, $i, j$ are indices on the plane's resolution. In the conditions of this task, we only set the MSE loss and the TV loss without adding any additional regularization in order to demonstrate the robustness of our method.

For the self-supervised decoupling of static and dynamic, the loss is:

$$\mathcal{L} = \mathcal{L}_c + \lambda_{tv}(\mathcal{L}_{tv}^{st} + \mathcal{L}_{tv}^{dy}) + \lambda_s \mathcal{L}_s + \lambda_d \mathcal{L}_d \tag{22}$$

$$\mathcal{L}_s(\mathbf{r}) = \frac{1}{N} \sum_{i=1}^{N} H\left( \frac{\sigma_i^{dy}}{\sigma_i^{dy} + \sigma_i^{st}} \right)^k \tag{23}$$

where $H(x) = -(x \cdot \log(x) + (1 - x) \cdot \log(1 - x))$

$$\mathcal{L}_d(\mathbf{r}) = \max\left( \frac{\sigma^{dy}}{\sigma^{st} + \sigma^{dy}} \right) \tag{24}$$

$\mathcal{L}_d$ and $\mathcal{L}_s$ are losses that help decouple scenes in D$^2$NeRF[28]. In fact, it can be seen in the framework that for the dynamic component part of it, the representation of the scene is higher than for the static part. Indeed, for the decoupling task in neural radiance field, dynamic components will inevitably have greater expressive power than static components.This can lead to the fact that during decoupling, dynamic components tend to incorrectly decouple the static in the scene as dynamic. In D$^2$NeRF[28] the authors control the shift of the extremes of $\mathcal{L}_s$ by adjusting the hyperparameter $k$. Thus, the dynamic part is suppressed and the density is preferentially decoupled to static. However, this parameter is not deterministic and needs to be readjusted for different scenarios to determine an optimal value, which makes the application of decoupling impractical.

Our goal is to achieve a more robust decoupling of the radiance field, so we no longer rely on the hyperparameter $k$. In all the experiments of the scenarios, we set it uniformly to 1.5. Instead we

propose a new optimization strategy, called different low-density initialization.

**Different low-density initialization.** For the decoupling settings, we propose to use different low-density initializations for static and dynamic components. Specifically, we use the *softplus* function to activate the density values, and add an additional offset $b$, for $\sigma = \text{softplus}(\sigma + b)$, we set the $b$ in the dynamic component much lower than the static, which can help to better decouple the dynamic from the static, and alleviate the dependence of self-supervision method on hyperparameter tuning.

**Coarse to fine training.** A coarse-to-fine sampling scheme is also used. Unlike [3, 5, 6], since we use the new four-dimensional coordinates after remapping, we progressively increase the resolution of all four dimensions. During training, we keep a mask to record low-density points in spacetime and skip these samples.

**Feature decoder.** For synthetic scenes, we use MLP as the density decoder; for real scenes, we directly sum the features to obtain the density. In both scenes, we use a small MLP as the appearance decoder.

**Windowed positional encoding.** We change the position coding to windowed position coding proposed in [19] in the decoupling task for real scenes. The window function is:

$$w_j(\alpha) = \frac{1 - \cos(\pi \operatorname{clamp}(\alpha - j, 0, 1))}{2} \tag{25}$$

where $j \in \{0, \ldots, m - 1\}$ is the index of the frequency band, and $\alpha \in [0, m]$ is linearly increasing. We use it for encoding the spatial coordinates before remapping and the appearance feature in the task of decoupling.

## 4 EXPERIMENTS

We experimented with our method on view synthesis and decoupling tasks in the monocular dataset. All experiments are implemented with PyTorch on an RTX 3090 GPU. The learning rate for feature planes is 0.02, and the learning rate for neural networks is 0.001 except for remapping net, which is 0.0007. All learning rates are exponentially decayed and kept in the same settings in all experiments. Please see supplementary material for visual comparisons of ablation experiments, detailed metrics and renderings.

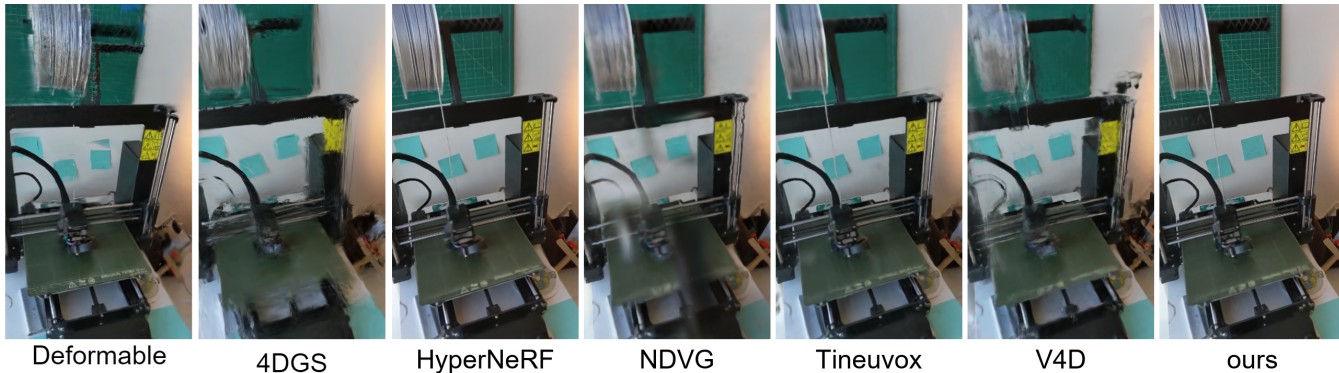

| Deformable 3DGS | 4DGS | HyperNeRF | NDVG | Tineuvox | V4D | ours |

**Figure 4: Qualitative comparison in real scenes**

## 4.1 Monocular Novel View Synthesis Results

For novel view synthesis, we used the 360° D-NeRF[21] dataset for synthetic scenes and the HyperNeRF[19] dataset for realistic scenes, respectively.

**D-NeRF dataset** is a monocular synthetic 360° dataset containing eight different dynamic scenes, each under large deformations and realistic non-Lambertian materials. The training set consists of 50 to 200 frames and the test set consists of 20 frames, each with a precise timing record.We set $F = 48$ for the appearance plane and $F = 24$ for the density plane,starts with $64^3$ for space resolution and $0.25 \times$ frames for time resolution and upsamples at 3k, 6k, 9k to $200^3$ and $0.5 \times$ frames,which is consistent with [3]. We show our results in Table 1, the results show that our method achieves better results in each metric compared to previous SOTA methods using this dataset including NDVG, V4D, Tineuvox, Hexplane, K-planes, D-NeRF and Temporal interpolation NeRF[3, 5–7, 9, 20, 21], and most recent method 4DGS[27], GS4D[31].See Supplementary Material for more detailed results and settings.

**HyperNeRF dataset** contains video sequences from mobile phones. There are four sequences of validation rig scenes that were recorded using two vertically aligned mobile phones. Unlike the D-NeRF dataset, the HyperNeRF dataset does not have a precise time record; it uses the sequential ids of the frames as time information. We set $F = 24$ for the appearance plane and $F = 8$ for the density plane, use summation instead of MLP to compute the density, and upsample resolution at 2k, 4k, 6k, and 8k. We show our results in Table 2, and please see Fig.4 for a visual comparison. We show in detail the relevant metrics and rendering pictures, which show that we have achieved better results. In addition we show the limitations of the current dynamic representation methods based on Gaussian splatting, which include deformable 3DGS[30] as well as 4DGS[27], these methods are not able to model rapid motions in the scene, such as the material columns of the printer, while our method can accurately reconstruct them. Please see more examples in supplementary material. It is worth mentioning that deformable 3DGS attributes this to the inaccuracy of the camera poses of the HyperNeRF dataset, this claim is inaccurate, since the static scene is reconstructed. Note that we do not and cannot use the deformation

**Table 1: Quantitative Results on D-NeRF dataset[21]**

| Method | PSNR ↑ | SSIM ↑ | LPIPS ↓ |
|---|---|---|---|
| Tineuvox[5] | 32.67 | 0.97 | 0.04 |
| NDVG[9] | 31.31 | 0.97 | 0.046 |
| V4D[7] | 33.72 | 0.98 | 0.02 |
| Hexplane[3] | 31.04 | 0.97 | 0.04 |
| K-planes-explicit[6] | 30.39 | 0.96 | - |
| K-planes-hybrid[6] | 30.84 | 0.96 | - |
| D-NeRF[21] | 30.50 | 0.95 | 0.07 |
| T-interpolation NeRF[20] | 32.73 | 0.97 | 0.03 |
| 4DGS[27] | 34.05 | 0.98 | 0.02 |
| GS4D[31] | 34.09 | 0.98 | - |
| ours w/o remapping | 31.20 | 0.97 | 0.04 |
| **ours** | **34.96** | **0.98** | **0.02** |

supervision provided in HyperNeRF[19] in the same way as other methods based on the deformation field.

## 4.2 Decoupling of Dynamic and Static Objects from a Monocular Video

We followed the dataset in $D^2NeRF[28]$, including some scenes in HyperNeRF[19] and new scenes. In keeping with the view synthesis settings, we test all of the vrig(validation rig) datasets include eight challenging real scenes. The corresponding experimental settings are consistent with view synthesis settings in real scenes. Due to the lack of relevant metrics to measure the effect of decoupling, we show the metrics for the synthetic result as well as a comparative picture of decoupling. The results show that our method takes the lead in both final synthesis metrics and training efficiency. Although we set the same hyperparameters for all scenes, we achieve comparable decoupling results,which proves the effectiveness of our strategies. In addition, the static backgrounds produced by $D^2NeRF$ always have noise in the occluded part of the dynamic object, which is not present in our method.

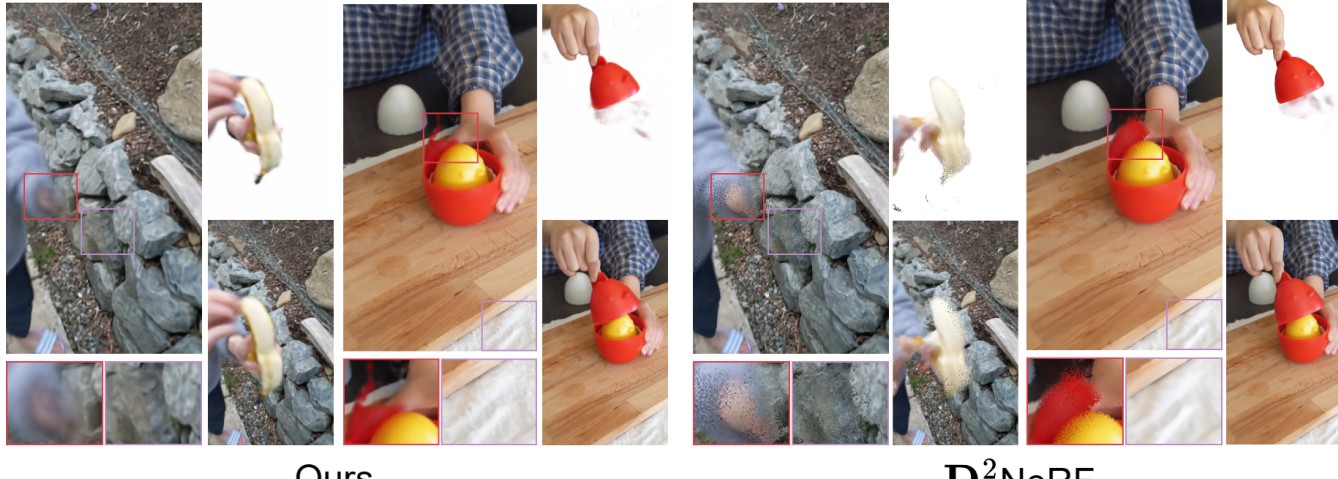

Ours                                    $D^2$NeRF

**Figure 5: Visual decoupling comparison of ours and $D^2$NeRF. In each part Left:static, Right top:dynamic, Right bottom: composite rendering. In static renderings, the decoupling region of $D^2$NeRF always contains dense noise, which is not present in our approach.**

**Table 2: Quantitative Results on HyperNeRF dataset[19]**

|          | V4D[7] | HyperNeRF[19] | NDVG[9] | Tineuvox[5] | T-interpolation[20] | 4DGS[27] | **ours** |
|----------|--------|---------------|---------|-------------|---------------------|----------|----------|
| PSNR ↑   | **24.8** | 22.4        | 23.3    | 24.3        | 24.35               | 25.2     | **25.28** |
| MS-SSIM ↑ | 0.832 | 0.814         | 0.823   | 0.837       | 0.866               | 0.845    | **0.871** |

**Table 3: Synthesis Results in the Decoupling Experiments**

|            | PSNR ↑ | MS-SSIM ↑ | Training Time ↓ |
|------------|--------|-----------|-----------------|
| **ours**   | **26.79** | **0.94** | **50 minutes**  |
| $D^2$NeRF[28] | 24.80 | 0.88     | 5~6 hours       |

## 4.3 Ablation Study

We performed ablation experiments on the remapping block on the D-NeRF dataset[21]. see Table 1, and please see Fig.6 for a visual comparison. The plane structure without remapping suffers from serious shortcomings in some monocular scenes.

For the decoupling task, we performed ablation experiments on the $D^2$NeRF dataset for the initialization strategy proposed in this paper. Without tuning the hyperparameters in the loss, our proposed initialization strategy significantly improves the effectiveness of decoupling. In contrast, the radiance field without initialization cannot be effectively decoupled in a self-supervised manner.

## 5 LIMITATION

Although our proposed remapping plane achieves good results in the tasks of novel view synthesis as well as dynamic and static decoupling, it still has shortcomings. In fact, since our method still follows volume rendering, compared to the current state-of-the-art 3D Gaussian splatting, for rendering speed, the 3D Gaussian splatting is much faster than the methods based on volume rendering,

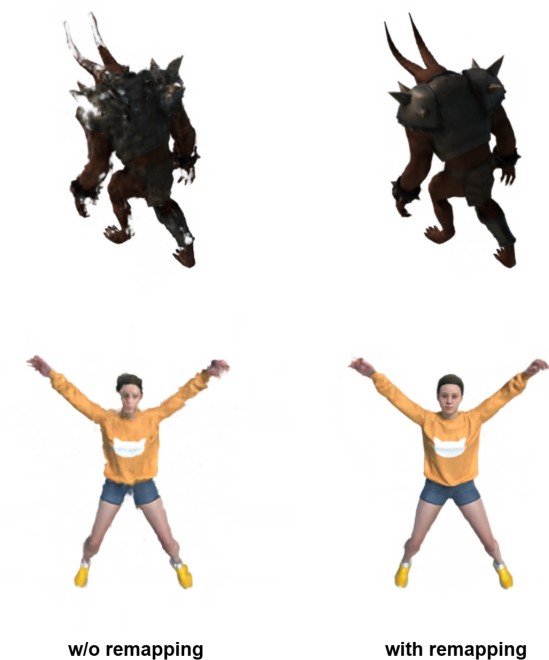

w/o remapping                    with remapping

**Figure 6: Ablation study of the remapping.**

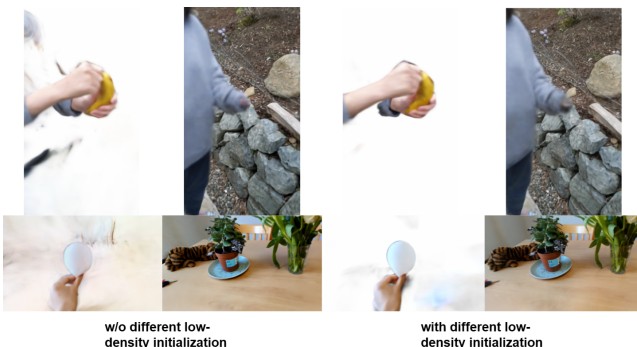

w/o different low-density initialization    with different low-density initialization

**Figure 7: ablations of the initialization in the decoupling task**

although in some scenes, our method is able to maintain comparable training speeds to the 3D Gaussian based representations. The trade-off between quality and efficiency remains an issue for subsequent researchers to consider.

## 6 CONCLUSION

We propose R4D-plane, which addresses the shortcomings of the plane structure by using the new 4D domain and its coordinates after remapping. In addition to view synthesis task, we extend it to dynamic-static decoupling. Experimental results of both tasks show the effectiveness of our approach, it significantly improves the rendering quality in monocular scenes and achieves faster and better dynamic-static decoupling compared to the previous approach. We hope that R4D-planes can contribute to the subsequent related work.

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
