# OpenReview forum: "R4D-planes: Remapping Planes For Novel View Synthesis and Self-Supervised Decoupling of Monocular Videos"
_acmmm.org/ACMMM/2024/Conference — MM2024 Poster_

### Official Review · Reviewer_K7eN · 2024-05-20

**Rating:** 4
**Confidence:** 3

**Summary:**

This article proposes a novel method for synthesizing dynamic scene views by decoupling dynamic static backgrounds and dynamic remapping. And based on tensor decomposition, it is beneficial to establish a compact and continuous time representation.  The experimental results also verified the dynamic and static decomposition ability of the model, and extensive experiments have proven the effectiveness of the method.

**Strengths:**

(1) The logic of the method seems to be correct, and the advantage of remapping has also been demonstrated in the ablation experiment.

(2) The explanation of the method is clear, and the relevant work citations are sufficient for me.

(3) The extensive efforts made in the experimental section are commendable. This effectively demonstrates the advantages of the proposed method.

**Limitations:**

(1) Although a large number of visual comparison results have been provided, dynamic synthesized videos have not been provided, whether from fixed or free viewpoints. Unable to further evaluate the quality of video synthesis.

(2)How to understand Line 331's "a mechanism for sharing information across frames" ? The model seems to have no cross-frame information exchange.

(3) The model seems to be an improvement on HexPlane, but the difference between the two is not reflected in the visualization results.

(4) Lack of comparison between training time and storage costs. Although decoupling training time is shown in Table 3, it does not seem to be the training time for the entire model. Qualitative analysis of the latest methods such as 4DGS, HexPlane, KPlane, etc. should be added, and these methods have already been publicly available in code.

**Suitability:**

2

---

### Official Review · Reviewer_riYg · 2024-05-23

**Rating:** 4
**Confidence:** 3

**Summary:**

This paper proposes a novel dynamic scene representation method based on tensor decomposition. By fusing spatio-temporal information and remapping to a new index, we compensate for the lack of planar structure. Experimental results show that R4D-planes outperform existing methods in terms of rendering quality and training efficiency for monocular scenes, as well as in dynamic and static decoupling tasks.

**Strengths:**

1. **High quality rendering results**. 3D scene rendering has always been a very important research field. The method proposed by R4D-planes improves the rendering quality and training efficiency, and has strong generalization. It can be applied to many fields such as augmented reality/virtual reality (AR/VR), movies and games, providing new technical means for 3D scene reconstruction and rendering in these fields.

2. **New processing method and loss function**. The authors propose a dynamic monocular reconstruction method based on remapping and a dynamic and static decoupling method based on tensor decomposition, and give the derivation formula and loss function in detail, which enhances the generalization of the paper.

3. **Extensive experiments.** Extensive experiments shows its effectiveness.

**Limitations:**

1. **Adaptability to complex scenarios:** Although this paper demonstrates excellent performance on specific datasets, the adaptability and robustness of R4D-planes in more complex and variable real-world scenarios (e.g., the ability to process complex motion patterns of dynamic objects) are not discussed in detail in this paper, and the adaptability and robustness of R4D-Planes need to be more extensively verified on different types of datasets.

2. **Limited ablation studies**. On the decoupling task, the influence of several proposed methods (such as different density initialization strategies, coarse-to-fine training schemes, decoder selection, and window position encoding) on the results is not explained in detail, which is expected to have a better qualitative representation.

3. **Complex methods design and background**. The proposed remapping and plane decoupling methods have a certain complexity in theory and implementation, which may increase the threshold of understanding and implementation. It is hoped that the introduction of the corresponding background knowledge, such as the definition of parameters and calculation methods, can be added in Section III.

**Suitability:**

2

---

### Official Review · Reviewer_ZUSH · 2024-05-24

**Rating:** 6
**Confidence:** 3

**Summary:**

This article introduces a novel approach for video novel view synthesis through the decoupling of static and dynamic objects using an Nd-Plan framework. The central innovation lies in the remapping function of the temporal dimension, which employs an activation function to distinguish between static and dynamic objects.

**Strengths:**

The proposed remapping extension improve the Nd-Plan NeRF framework in a straightforward and logical way by reducing the complexity of space-time representation with the mapping function without adding extra "hack". The article is clearly written and very easy to understand thanks to the relevance of the idea. The improvement in the evaluation is good enough to demonstrate the impact of the remapping function when using Nd-Plan framwork.

**Limitations:**

While the limitations discussed in the article are valid, there are additional concerns. This method may perform poorly in scenarios where objects move slowly in the whole scene or have low velocity, such as a tree swaying in the wind. the remapping function might have drawbacks in image rendering when compared to the traditional 4D K-Plan approach. These cases should be addressed by the authors in someway (adding an extra evaluation or a short paragraph on it). Additionally, the XZ plane splitting in Figure 1 is unclear for me.

**Suitability:**

3

---

### Official Review · Reviewer_HBnS · 2024-05-26

**Rating:** 3
**Confidence:** 4

**Summary:**

The paper introduces R4D-planes, a new representation for dynamic scenes based on tensor decomposition aimed at improving novel view synthesis and self-supervised decoupling of monocular videos. The proposed method addresses the limitations of previous NeRF-based methods and explicit representations by introducing a remapping technique that fuses space-time information and compensates for the shortcomings of the plane structure. Additionally, a new decoupling structure is implemented to efficiently separate dynamic and static scenes. Experimental results indicate that R4D-planes achieve better rendering quality and training efficiency in both tasks compared to existing methods.

**Strengths:**

1. The paper tackles important challenges in computer vision and computer graphics, specifically in the areas of view synthesis and dynamic scene decoupling.
2. Experiments on two datasets demonstrate the superiority of the proposed method.

**Limitations:**

1. The paper lacks experiments on the Plenoptic Video dataset, which is a significant omission given the relevance of this dataset to the tasks at hand. Including results from this dataset would provide a more comprehensive evaluation of the proposed method's performance.
2. There is a lack of detailed ablation studies to demonstrate the effectiveness of the remapping technique. The paper should also provide a thorough analysis of why it improves performance. What limitations of the previous methods that can be addressed by the remapping technique are not clear.
3. From Figure 4, it is difficult to judge which one is better, HyperNeRF or the method proposed in this paper. The authors are encouraged to zoom in on the details so that readers can quickly focus on the distinguished areas.
4. The authors claim that deformable 3DGS and 4DGS cannot model rapid motions in the scene, while the proposed method can accurately reconstruct them. However, one cannot make such a conclusion from the Figure 4 in the sumplementary materials. Besides, if a video demo can be provided in the sumplement, it would be easier for readers to observe the superiority of the proposed method.
5. In Table 2, the PSNR of V4D should not be boldfaced.

**Suitability:**

3

---

### Meta-Review · Area_Chair_Rfrm · 2024-06-28

**Recommendation:** Accept (Poster)
**Confidence:** 4

**Metareview:**

All the reviews were positive originally. The reviewers found that this paper is well written, the idea is sound and the improvement is good to demonstrate the impact of the remapping function when using Nd-Plan framework. After the rebuttal, one of the reviewers lowered the score from borderline acceptance to borderline reject due to the concern with the lack of significant advantage in computational cost, the presentation in Table 2 and the insufficient evidence of good motion reconstruction quality. Given the overall positive reviews and most of the concerns raised by reviewer K7eN can be addressed in the revision, we'd like to recommend the acceptance of this paper. We would encourage the authors to carefully revise the paper to incorporate reviews to improve the paper.